# The Implication of Chemotypic Variation on the Anti-Oxidant and Anti-Cancer Activities of *Sutherlandia frutescens* (L.) R.Br. (Fabaceae) from Different Geographic Locations

**DOI:** 10.3390/antiox9020152

**Published:** 2020-02-13

**Authors:** Samkele Zonyane, Olaniyi A. Fawole, Chris la Grange, Maria A. Stander, Umezuruike L. Opara, Nokwanda P. Makunga

**Affiliations:** 1Department of Botany and Zoology, Stellenbosch University, Private Bag X1, Merriman Avenue, Stellenbosch 7602, South Africa; 15965678@sun.ac.za (S.Z.); chrislag@sun.ac.za (C.l.G.); 2South African Research Chair in Postharvest Technology, Department of Horticultural Science, Stellenbosch University, Private Bag X1, Merriman Avenue, Stellenbosch 7602, South Africa; olaniyi@sun.ac.za (O.A.F.); opara@sun.ac.za (U.L.O.); 3Department of Botany and Plant Biotechnology, University of Johannesburg, P.O. Box 524, Auckland Park, Johannesburg 2006, South Africa; 4Central Analytical Facility, Stellenbosch University, Private Bag X1, Merriman Avenue, Stellenbosch 7602, South Africa; lcms@sun.ac.za

**Keywords:** cycloartane glycoside, flavonoids, LC-MS, metabolite profiling, terpene saponins, sutherlandioside, sutherlandin, cancer bush

## Abstract

Extracts of *Sutherlandia frutescens* (cancer bush) exhibit considerable qualitative and quantitative chemical variability depending on their natural wild origins. The purpose of this study was thus to determine bioactivity of extracts from different regions using in vitro antioxidant and anti-cancer assays. Extracts of the species are complex and are predominantly composed of a species-specific set of triterpene saponins (cycloartanol glycosides), the sutherlandiosides, and flavonoids (quercetin and kaempferol glycosides), the sutherlandins. For the Folin-Ciocalteu phenolics test values of 93.311 to 125.330 mg GAE/g DE were obtained. The flavonoids ranged from 54.831 to 66.073 mg CE/g DE using the aluminum chloride assay. Extracts from different sites were also assayed using the 2,2-diphenyl-1-picrylhydrazyl (DPPH^•^) radical scavenging method and ferric reducing anti-oxidant power (FRAP) methods. This was followed by an in vitro Cell Titer-Glo viability assay of various ecotypes using the DLD-1 colon cancer cell line. All test extracts displayed anti-oxidant activity through the DPPH^•^ radical scavenging mechanism, with IC_50_ values ranging from 3.171 to 7.707 µg·mL^−1^. However, the degree of anti-oxidant effects differed on a chemotypic basis with coastal plants from Gansbaai and Pearly Beach (Western Cape) exhibiting superior activity whereas the Victoria West inland group from the Northern Cape, consistently showed the weakest anti-oxidant activity for both the DPPH^•^ and FRAP methods. All extracts showed cytotoxicity on DLD-1 colon cancer cells at the test concentration of 200 µg·mL^−1^ but *Sutherlandia* plants from Colesburg (Northern Cape) exhibited the highest anti-cancer activity. These findings confirm that *S. frutescens* specimens display variability in their bioactive capacities based on their natural location, illustrating the importance of choosing relevant ecotypes for medicinal purposes.

## 1. Introduction

*Sutherlandia frutescens* (L.) (=*Lessertia frutescens* (L.) R.Br. (Fabaceae) has a complex chemistry that contains a unique set of flavonoids and triterpene saponins amongst other chemical constituents [1]. Flavonoids, a phytochemical group of polyphenolic compounds, are recognized for health benefiting effects exhibiting a diverse range of biological activities such as anti-oxidant, anti-inflammatory, antimicrobial, anti-tumoral, anti-thrombogenic, anti-therosclerotic, anti-viral, and anti-allergic properties; to name a few (for details refer to Umesh et al. [2]). They are some of the most effective anti-oxidant compounds available to humans. They exert their anti-oxidant effect by scavenging oxygen derived free radicals which are known to have adverse effects on health that lead to the development of various degenerative diseases such as cancers, brain dysfunction, cardiovascular diseases, and also, those that are associated with a weakened immune system [2]. The polyphenols, in general, act by inhibiting or delaying the initiation of oxidative chain reactions. Therefore, the presence of flavonoids creates more stable and less reactive radicals when the hydroxyl group in these phytochemicals have reacted and been oxidized by radicals themselves [3]. Flavonoids are also known to exhibit anti-oxidant action by chelating iron which causes peroxidation when combined with reactive oxygen species [2]. These phytochemical constituents also directly inhibit lipid peroxidation that affects the integrity of lipid membranes of cells, which is another measure for protection from the adverse effects of oxidation [2,3]. The production of anti-oxidants in plants is ubiquitous in nature, but those plants with specialized metabolites that are species specific, may have better anti-oxidant effects than other species.

Triterpene saponins have in the recent decade become very interesting as drug candidates for cancer chemotherapy due to their cytotoxic and cytostatic effects. The work of Thakur et al. [4] reviews the potential of these chemicals as anti-cancer agents as triterpene saponins may exhibit their action by either slowing the growth of tumor cells or result in cell death. These actions are linked to the permeabilization of membranes, alterations to cellular anion channel function, mitochondrial and endoplasmic reticulum (ER) malfunction, and stimulation of immune responses in the presence of carcinogens [4]. Furthermore, they may protect DNA from damage and thus act as anti-mutagens. As a result, plants with unique triterpenoidal saponins are highly sought after as potential sources of new anti-tumorigenic drugs.

Scientific interest in *S. frutescens* is also on the rise as its extracts have been shown to exhibit anti-cancer activity in in vitro assays. It is important to recognize that, at this stage, our knowledge on the phytochemical composition of *S. frutescens* is still shallow as not all metabolites that are synthesized by this species have been completely elucidated. Progress in this regard dates back to the late 1960s when triterpene saponins in *S. frutescens* were first reported [1], but it was only in the past decade, that chemical biomarkers were purified (refer to Appendix A for chemical structures). Specifically, the triterpene saponins of Sutherlandia, (as it is often called in colloquial terms) are in fact cycloartane glycosides, termed sutherlandiosides (Appendix A). These are characteristic of the species and they are known to show site-specific variation [5]. They are nowadays used in quality control protocols as part of the regulatory measures in a commercial production pipeline [5]. Thus far, the flavonoids that have been characterized in *S. frutescens* are those with a kaempferol or quercetin aglycone (Appendix A) and these flavonoid glycosides are also thought to be important contributors to the various pharmacological activities of the species. Since the bioactivity of medicinal plants is ascribed to the presence of chemical compounds contained by the plant, the chemotypic variation of *S. frutescens* (shown by Albrecht et al. [5] and Zonyane et al. [6] in previous work) might cause variable bioactivity of plants collected from differing wild habitats. There is a link between strong anti-oxidant activity and chemopreventative anti-tumor actions mediated by herbal products but the efficacy of the herbal preparation is often determined by the quality and source of the plant [5].

Although the functional relevance of sutherlandins and sutherlandiosides in the plant is not yet known, plants in general, are phenotypically plastic, responding to micro-environmental changes by adjusting metabolic flux leading to chemical-based variation [6]. *S. frutescens* is widely distributed in South Africa, occurring in different biomes and where it is found, it is an important herbal medicine that is used by various cultural groups and its record of use spans across centuries [1]. The plant is popular for the treatment of a wide range of ailments including stomach diseases, stress, fever, wounds, diabetes, and cancers, hence, the Afrikaans name of ‘kankerbos’ or cancer bush in English [1]. Because this species has a strong reputation in folklore as a medicinal plant that treats cancers, studies to validate these claims are plenty. For example, some studies showed that extracts lead to apoptosis of breast cancer cell lines with MCF-7 tumor cells being more susceptible to cell death in comparison to the MCF-12 A cells [7,8]. They also inhibit growth of CaCo_2_ colon cancer cells by disrupting cell membranes and cell death is mediated via the P13-K pathway [9]. Various publications also indicate the anti-oxidant properties of *S. frutescens* [10,11,12,13] but at this stage, the key bioactive(s) that may hold anti-cancer activity are not yet known. Furthermore, there is limited information regarding specific ecotypes and their anti-cancer activities. Most studies typically use material from one chosen natural population or material that has been farmed but without provenance information being provided in detail [6]. The latest research shows the cytotoxicity effects of both water and ethanolic extracts of *S. frutescens* using a zebrafish bioassay and this effect is clearly dose-dependent [14]. Other recent studies have tried to explain the mechanisms that may dictate the physiological effects of various extracts of *S. frutescens* and the NF-kB, MAPK, and TNF signaling are linked to immune-modulation effects [15]. Ntuli et al. [16] showed anti-mutagenic effects of ethyl acetate and methanol extracts using an AMES analysis. Several authors use commercially available products of *Sutherlandia* to obtain plant material for pharmacological testing (see work of Leisching et al. [9]; Ntuli et al. [16]; Lin et al., [17]; and Van Der Walt et al. [18]). However, the manufacturers of these products do not necessarily attest to the origins of the chemotypes used in the manufacturing process and neither do they indicate where the mother stock plants used for building an agricultural crop of the species actually came from.

To add to a growing body of knowledge with regards to *S. frutescens* and its chemotypes, the aim of this particular study was two-fold. At first, it was to establish whether chemical variation among *S. frutescens* plants from different places has any implications on anti-oxidant activity of this species because these plants possess flavonoids and terpene saponins in the form of sutherlandins and sutherlandiosides, respectively. All other studies focused on using either wild or field-grown plant material to validate the effects of *S. frutescens* extracts in bioassays, and in some instances, commercially purchased leaves without taking into consideration the origins of the plant material have been used as test materials. The second aim was, therefore, to test for in vitro anti-cancer effects using a DLD-1 colon cancer cell line which has not been previously tested; and, to establish which of the wild chemotypes identified by Zonyane et al. [5] showed the greatest potency as accumulation of the sutherlandins and sutherlandiosides is site specific. This paper thus adds additional evidence to support the popularity of this medicinal plant as an anti-cancer agent and directly correlates both anti-oxidant and anti-cancer bioactivities to inter-population based chemical variation.

## 2. Materials and Methods

### 2.1. Plant Collection and Extraction

*Sutherlandia frutescens* plants growing in the wild were collected from various populations in different provinces of South Africa (Table 1). We targeted the month of November 2014 for these collections as this is the flowering season. The characteristic orange-red flowers were used as a diagnostic feature to identify plants in the wild. Most plant material was collected by SZ (representative voucher numbers: SF (Z14 to 19)) as only one set of samples, from Pearly Beach, were collected by NPM (NM(1)). A set of voucher specimens was lodged with the herbarium of the Department of Botany and Zoology (Stellenbosch University, Stellenbosch, South Africa) and during this time, their botanical identity was confirmed.

Plant material was dried at room temperature and stored in the dark for the duration of the study. Since plants from the same habitats displayed similar LC-MS profiles based on previous work from our laboratories (see also Zonyane et al. [6], leaf material from four individual plants from each of the seven localities were pooled together for the anti-oxidant and anti-cancer analyses (Table 1). One gram of dried and pulverized leaves of *S. frutescens* was extracted with 20 mL of 75% (*v*/*v*) ethanol and an ultra-sound sonicator (Branson B220H, Danbury; United States of America). for 60 min was used to increase extraction efficiency. The distilled water in the sonicator was kept cool by the addition of ice during this period. After the extraction, cell debris was removed from the plant extracts using Whatman^®^ No. 1 filter paper (Merck, Darmstadt, Germany). The filtered extracts were then concentrated to dryness by evaporating the solvent at room temperature in the dark for 72 h. Extractions were conducted in triplicate. Each dried extract was weighed and resuspended with 75% (*v*/*v*) ethanol to obtain a stock extract with a concentration of 1 mg·mL^−1^.

### 2.2. Measurement of the Total Phenolic and Flavonoid Content

The total phenolic concentration of various *S. frutescens* plants representing each study collection site was determined spectrophotometrically by the Folin–Ciocalteu method [19] whereby 50 µL of the plant extract was mixed with 450 µL of 50% (*v*/*v*) methanol and 500 µL of Folin–Ciocalteu reagent. After 2 min, 2.5 mL of 2% (*v*/*v*) sodium carbonate was added to the mixture and the contents were vortexed to mix thoroughly. The reactions were kept at room temperature in the dark for 40 min, after which the absorbances were measured at the wavelength of 725 nm with a UV-Vis spectrophotometer (Thermo Scientific Technologies, Madison, Wisconsin). The total phenolic content of various *Sutherlandia* samples was quantified by a calibration curve that was obtained from absorbance readings of known concentrations of gallic acid standard solutions (2 µg·mL^−1^ to 16 µg·mL^−1^ resuspended in 50% methanol; A = 0.02 c (gallic acid) − 0.0286, *R*^2^ = 0.9941). This experiment was conducted in triplicate. The data were expressed as mg of gallic acid equivalent per grams of dry extract (mg GAE/g DE).

To determine the total flavonoids content of *S. frutescens* samples, an aluminum chloride colorimetric assay was used in which 250 µL of the plant extract was mixed with 1.25 mL of distilled water and 75 µL of 5% (*v*/*v*) sodium nitrite. Samples were left for 5 min before 150 µL of 10% (*v*/*v*) aluminum chloride solution was added. This was followed by addition of 500 µL of 1 M sodium hydroxide solution and 775 µL of distilled water after 5 min. The absorbances were measured immediately thereafter at 510 nm using a spectrophotometer (Thermo Scientific Technologies, Madison, WI, USA). The total flavonoid concentration of samples was calculated using the calibration curve of catechin (10 µg·mL^−1^ to 500 µg·mL^−1^ resuspended in 50% methanol; A = 0.0059 c(catechin) + 0.6245, *R*^2^ = 0.8592). The assay was conducted in triplicate. The data were calculated to represent mg of catechin equivalent per gram of dry extract (mg CE/g DE).

### 2.3. Anti-Oxidant Activity

The anti-oxidant activity was assessed by two methods. The first method involved using the free radical scavenging method based on the 2,2-diphenyl-1-picrylhydrazyl (DPPH^•^) assay which measures the hydrogen donating ability of the plant extract by reducing DPPH^•^ to DPPH-H; and, this is physically indicated by the bleaching of a purple-colored DPPH^•^ solution. The ferric reducing anti-oxidant power (FRAP) assay which involves the reduction of a ferric-tripyridyltriazine complex to its ferrous colored form in the presence of anti-oxidants was also used.

#### 2.3.1. DPPH^•^ Radical Scavenging Activity

To measure the anti-oxidant activity using DPPH^•^ method, 15 µL of the plant extract was mixed with 735 µL of methanol and 750 µL of 0.1 mM DPPH^•^ solution. The reaction contents were mixed thoroughly using a vortex and incubated at room temperature for 30 min and covered with aluminum foil, after which the absorbances of the reaction solutions were measured at the wavelength of 517 nm. Even though DPPH^•^ is a relatively stable free radical, it gradually deteriorates when it is in solution form, hence, the assay was conducted under dark conditions and only freshly prepared solutions were used. An anti-oxidant, ascorbic acid was used as a positive control (0.4 mM to 2.4 mM resuspended in methanol); A = −0.298c (ascorbic acid) + 0.7482, *R*^2^ = 0.9843. The free radical scavenging activity (RSA) as determined by the decoloration of the DPPH solution was calculated using the following formula:Scavenging (%) of DPPH = (1 − A_E_)/A_D_ × 100
where: A_E_ = *Absorbance of the reacted extract or antioxidant standard*; A_D_ = *absorbance of the DPPH solution only*.

A dilution series (1.25, 2.50, 5.00, 10.0, 20.0, and 40 µg·mL^−1^) for each of the samples from different sites was prepared to give approximately 0% to 100% inhibition of DPPH^•^. The radical scavenging capacity of *S. frutescens* extracts was then evaluated in terms of their respective IC_50_ values (inhibitory concentration of the extract (µg) necessary to neutralize DPPH by 50%) with lower IC_50_ values indicating higher hydrogen donating (anti-oxidant activity) ability and vice versa.

#### 2.3.2. Ferric Reducing Anti-Oxidant Power (FRAP) Assay

To measure the anti-oxidant activity using the FRAP method, 150 µL of the plant extract was added to 2850 µL of the FRAP solution which contained 300 mM sodium acetate buffer (pH 3.6), 10 mM 2,4,6-Tri(2-pyridyl)-s-triazine (TPTZ) solution and 20 mM ferric chloride. The reaction contents were thoroughly mixed and incubated for 30 min at room temperature in the dark. The absorbances were measured, thereafter, at a wavelength of 593 nm. Trolox (0.1 to 1 mM resuspended in 50% methanol; A = 1.8872c (Trolox) + 0.3194, *R*^2^ = 0.9631) was used as the standard. The anti-oxidant activity was expressed as mM Trolox equivalent per gram of dry extract (mM TE/g DE).

### 2.4. Anti-Cancer Activity

#### 2.4.1. Cell Culture

The anti-proliferative effect of *S. frutescens* specimens was evaluated on a colon adenocarcinoma cell line (DLD-1; American Type Culture Collection) as at the time of designing these experiments, the *S. frutescens* extract had not previously been tested on this cell line. The DLD-1 cells were provided by Reaction Biology Corporation where tests were performed (Pennsylvania, USA). The DLD-1 cells were cultured in 20 µL of RPMI-1640 medium which was supplemented with 10% (*v*/*v*) fetal bovine serum, 100 µg/mL of penicillin and 100 µg/mL of streptomycin. Cultures were maintained at 37 °C in a humidified atmosphere with 5% CO_2_ (*v*/*v*) and 95% air (*v*/*v*) and to prevent liquid evaporation of the medium, a porous microplate sealing film was used to seal the 383-well plate during the cell incubation period.

#### 2.4.2. Cell Titer-Glo Viability Assay

To conduct the Cell Titer-Glo Viability Assay, the cancer cells were seeded into wells of the 384-well cell culture plate at 2000 cells per well in triplicate. This was followed by incubation at 37 °C in 5% CO_2_ overnight to mediate attachment of cells to the wells. Then, each plant extract was added to the wells of a plate as 5 µL aliquots and the plates were incubated for 48 h for the duration of treatment. Plant extracts were analyzed in a 10-dose-dependent manner with three-fold serial dilutions and subsequently, the concentrations that were tested were; 0.01, 0.03, 0.10, 0.30, 0.80, 2.50, 7.40, 22.2, 66.7, and 200 µg·mL^−1^ for their cytotoxicity effects on test cancer cells. A protein-kinase inhibitor, 5 µL of staurosporine (Sigma-Aldrich, Michigan), at the concentration of 10 µM, was used as the reference compound and was diluted and screened in the same manner as the plant extracts. After 48 h, 25 µL of CellTiter-Glo^®^ 2.0 luminescent reagent (Promega (Madison, WI, USA)) was added to each well. The contents were mixed on an orbital shaker for 2 min and incubated at room temperature for 15 min to stabilize the luminescent signal. Luminescence was recorded by Envision 2104 Multilabel Reader (PerkinElmer, Santa Clara, CA, USA). The number of viable cells in culture was determined based on quantitation of the ATP present in each culture well. This assay was performed at Reaction Biology Corporation (PA, USA).

### 2.5. Phytochemical Profiling

A Waters Acquity Ultra Performance Liquid Chromatography (Milford, MA, USA) coupled with an Acquity photo diode array (PDA) detector and a Waters Synapt G2 quadrupole time-of-flight mass spectrometry was used to analyze phenolic and terpenoid compounds in *S. frutescens* samples. Using an autosampler, 1 μL was injected for analysis. The chromatography was performed using a Waters UPLC BEH C18 column (2.1 mm × 100 mm, 1.7 μm particle size) under a flow rate of 350 μL min^−1^ for 26 min. Water (A) and acetonitrile (B) (both containing 0.1% formic acid) were combined for the elution and the gradient elution was performed as follows: 5% B at 0.5 min, 44% B at 20 min, 100% B at 21 min, 100% B at 22 min, 5% B at 23 min, and A5% B at 26 min. Other conditions were set at 15 V for the cone voltage, 3.0 kV for the capillary voltage. Data were acquired in MS^E^ mode and a collision energy of 6 V (low energy transition) to obtain MS data, with a ramp setting of 15 to 60 V for the trap collision energy to obtain fragmentation data. This protocol uses nitrogen as the desolvation gas (650 L h^−1^ flow rate and at 275 °C) and a positive electrospray ionization mode. The flavonoids (sutherlandins) and terpenoids (sutherlandiosides) eluted between 6.3–9 min and 15.9–21.6 min, respectively. All spectra of the largest peaks and were tabulated in an excel spreadsheet. This was followed by combining the mass spectra data, lambda (UV) and fragmentation (MS^E^) data to make putative chemical assignments.

### 2.6. Data Analysis

All the anti-oxidant data are presented as mean including the standard deviation (mean ± SD). The results were compared by one-way analysis of variance (ANOVA) to determine significant differences between the means after normality testing. Differences that are at *p* < 0.05 between the means were considered statistically significant. The IC_50_ values for DPPH reduction were calculated on GraphPad Prism 7.04, within the dilution series that was prepared.

The in vitro anti-cancer assay results were interpreted in terms of percentage cellular viability which was expressed as mean ± SD. The Tukey post-hoc test was used to compare which means were significant from each other at the significance level of *p* < 0.05. Plant extract concentrations at which 50% cell growth is inhibited (IC_50_) were determined by the log sigmoid dose-response curve plotted on GraphPad Prism 4. Data were utilized to construct a principal component analysis illustrated as a biplot and a Pearson’s correlation matrix was used to visualize relationships between polyphenolic content, anti-oxidant-, and anti-cancer activities.

## 3. Results

### 3.1. Anti-Oxidant Analyses

It is well established that all plants possess anti-oxidant activity but variations in different phenolic structural profiles lead to different anti-oxidant capacities for plant extracts and extracts of *S. frutescens* from different areas were variable (Figure 1).

Both qualitative and quantitative differences seem to be expressed as a result of geographic location. Total phenolic content in plants is largely implicated in their anti-oxidant power and ethanol extracts of various specimens of *S. frutescens* obtained from different geographic regions varied in their composition of total phenolics and flavonoids (Figure 2).

The content of phenolics ranged from 93.311 to 125.330 mg GAE/g DE with plants from Gansbaai 1 (Western Cape) having the highest concentration followed consecutively by plants from the nearby region of Pearly Beach (Western Cape) (Figure 2A). Of interest is that, all the Western Cape plants studied in this particular investigation have been shown to be a chemotype that accumulates sutherlandin B (Table 1). *Sutherlandia* plants from Burgersdorp (Eastern Cape) had the lowest value of phenolics measured (Figure 2A), and predominantly, these plants accumulate the cycloartenol glycosides, sutherlandioside A and sutherlandioside B (Table 1). The phenolic concentration of plants from the other wild collection sites were relatively similar and generally significantly lower than the levels of phenolics that were recorded for *Sutherlandia* plants from the Western Cape group (*p* < 0.05) (Figure 2A). The total flavonoid values are displayed in Figure 2B and flavonoid constituents ranged from 54.831 to 66.073 mg CE/g DE. *Sutherlandia* plants from Pearly Beach had the highest composition of flavonoids recorded with the in vitro assay. Similarly, to the phenolic analysis, the levels of flavonoids recorded for studied specimens followed the same trend with Western Cape having significantly higher amounts as opposed to those plants from other provinces.

The measurement of the total anti-oxidant phytochemicals was followed by an assessment of the anti-oxidant capacity of various *S. frutescens* specimens, shown in Figure 2C,D. As expected, all the samples from different habitats displayed anti-oxidant activity as demonstrated by both DPPH^•^ and FRAP assays that were used, and IC_50_ values ranged between 3.171 and 7.707 µg·mL^−1^ (Figure 2C). The plants from Victoria West displayed an IC_50_ value of 7.707 µg·mL^−1^, thereby indicating the lowest radical scavenging activity recorded among all studied specimens of *S*. *frutescens*, whereas those plants from the Pearly Beach subset had the strongest bioactivity (Figure 2C). Figure 2E shows the IC_50_ curve for free radical (DPPH^•^) scavenging activity and these data further emphasize the higher DPPH^•^ power of the Gansbaai 1 population group. Both DPPH^•^ and FRAP methods showed a strong correlation between the total phenolic and flavonoid content and the degree of anti-oxidant activity of *Sutherlandia* plants with *r*^2^ values of 0.9032 and 0.9163 for radical scavenging activity, respectively, and 0.9501 and 0.9834 for reducing anti-oxidant activity, respectively (Figure 3).

Again, all *S**. frutescens* specimens from various localities displayed anti-oxidant activity when they were assessed for reducing anti-oxidant power through FRAP assay. Plants from Gansbaai 1 and Pearly Beach exhibited equal and highest reducing anti-oxidant powers at 16.2319 mM TE/g DE, followed by plants from Burgerdorp and Colesburg, both with the reducing anti-oxidant powers at 15.4162 mM TE/g DE (Figure 2D). The two methods, DPPH^•^, and FRAP assays, used to assess anti-oxidant activity in this study were more congruent with each other (Figure 2 and Figure 3) and therefore able to conclusively show those *S. frutescens* specimens that were more potent in their bioactivities. Both methods confirmed the presence of variability in the anti-oxidant activities among wild-collected plants and this was largely correlated to biogeographic zones where plants were harvested from and those plants from coastal localities being more powerful.

### 3.2. Anti-Cancer Activity

*S. frutescens* is reputed as an anti-cancer agent and we were thus interested to examine the possibilities of variable anti-proliferation effects linked to extracts from disparate habitats. This idea was based on the fact that there is no consistency and congruency in recorded data as various laboratories will only collect material from one site to conduct their investigations. The amount of ATP, as measured by Cell Titer-Glo viability assay was used as a signal for cellular metabolic activity, therefore, acting as an indirect measure of cell death in extract treated cancer cells. At the highest test concentration of 200 µg·mL^−1^, all tested extracts demonstrated cytotoxicity on test colon cancer cells (DLD-1).

However, specimens had varying degrees of cytotoxicity on the cancer cells, with plants from Colesburg (Northern Cape) exhibiting the highest anti-cancer activity (36.6% viability). Even though, Victoria West is in the Northern Cape (63.2% viability) where some studies have shown plants from this area to have superior anti-tumor effects [5], these plants were least effective in the anti-cancer activity test in comparison to extracts produced from other localities (Figure 4). There was no particular trend connecting the anti-cancer activities with anti-oxidant activities in Section 3.1, except plants from Gansbaai 1 which showed consistency in their relatively higher anti-oxidant and anti-cancer activities (Table 2). Between concentrations of 22.2 µg·mL^−1^ and 200 µg·mL^−1^, all the plant extracts induced cytotoxicity on DLD-1 cells in a dose-dependent manner illustrating a relationship regarding the concentration and cytotoxicity effects where higher concentrations of the extract prevented the growth of the cancer cells. Concentrations lower than 22.2 µg·mL^−1^ which ranged from 0.01 µg·mL^−1^ to 7.4 µg·mL^−1^ induced similar cytotoxicity on DLD-1 cells (*p* > 0.05). The IC_50_ values further confirmed the relatively higher potency of Colesburg, Zastron and Gansbaai 1 plants as specimens from these regions had IC_50_ values of 158.7, 172.7, and 176.7 µg·mL^−1^, respectively (Table 2).

### 3.3. Correlation Matrix and Principal Component Analysis

A Pearson’s correlation matrix was used to reveal the degree of correlation between two parameters, confirming relationships between the test indices (Figure 5A). As expected, significant positive correlations were recorded for total phenolics and flavonoid content (*r* = 0.916, *p* = 0.05) but the DPPH activity was negatively correlated to the flavonoid and total phenolic content (*r*= −0.847 and *r* = −0,872 at the *p* = 0.05 level of significance, respectively). The highest correlation efficient of 0.985 was recorded for the IC_50_ values versus the anti-cancer activity and this was also statistically significant.

The biplot with resulting PC 1 and PC 2 axes showing principal components corresponding to 68.5% and 22.6% of the variation, respectively, was constructed to further correlate bioactivity and biogeographic origins of the extracts (Figure 5B). In respect to the overall bioactivity, the biplot shows the separation of *S. frutescens* plants, from various regions, into three main clusters. The first cluster has plants from Colesburg and Zastron which had remarkable high anti-proliferative activity against DLD-1 cancer cells. On the other hand, the second cluster is composed of *S. frutescens* plants from Victoria West, Burgersdorp and Gansbaai 2, characterized by high DPPH IC_50_ values which signify relatively low anti-oxidant activity. The *S. frutescens* plants in the third cluster were from Gansbaai 1 and Pearly Beach and the plants from these regions were mainly characterized by high FRAP activity and high contents for the presence of flavonoids and phenolics. These plants further had an attribute of low IC_50_ values for DPPH activity signifying the high anti-oxidant activity they possessed.

The largest peaks, presented in Figure 1, were used to characterize the profiles of the major chemicals present in *Sutherlandia* plants across all populations (Table 3). It is clear that extracts of *S. frutescens* are complex with high variation quantitatively in their pool of compounds with anti-oxidant power. In this study, there are several other kaempferol and quercetin derivatives that were present in the extracts that have not been fully characterized and remain unknown. The bioactives that have anti-cancer activity thus need to be explored further. However, it is clear that anti-cancer activity shown here is not necessarily strongly correlated to the flavonoids. Some of the other chemicals that we were able to tentatively identify include isomers of the sutherlandins, some triterpenoids and cycloartanol glycosides (sutherlandiosides). The cycloartanol glycosides have characteristic MS spectra showing characteristic cycloartanol aglycone fragments (Appendix A), a low double bond equivalency and no UV peaks [5]. The sutherlandins are flavonols with characteristic UV spectra with a UV max at 265 and 345 nm and also a base peak in the higher energy trace that corresponds to the specific aglycone [M+H], eg *m*/*z* 303.05 for quercetin and *m*/*z* 287.055 for kaempferol (Appendix A).

## 4. Discussion

The Western Cape samples generally had higher levels of the anti-oxidant phytochemicals and displayed higher anti-oxidant activities compared to the plants from other provinces within the Karoo region (Figure 2 and Figure 3). Even though the Western Cape plants performed better than plants from other provinces, they also had considerable variation among themselves with plants from Gansbaai 2 consistently being the least active, whereas the plants from the Karoo region had anti-oxidant phytochemicals (phenolics and flavonoids) that were relatively uniform among each other (Figure 1A,B). In fact, Karoo specimens have a distinct metabolomic profile that is constituted of the major triterpene saponin, sutherlandioside B [14]. This is interesting as there has been a perception, since the isolation and purification of this chemical, that it could play a strong role in the cytotoxicity actions of *Sutherlandia* plants and it may have other pharmacological functions, hence it is monitored in products manufactured from the raw materials of this species [5]). Recently, the study of Lin et al. [17] showed those fractions with sutherlandioside D to have high anti-tumorogenic effects against prosrate cancer using an in vivo setup suggesting that these anti-cancer effects are exerted through Gli/Hn signaling pathways that control cell patterning and formation plus proliferation in animal cells. A previous study identified a sutherlandioside D isomer in plants obtained from the Gansbaai area [5] but plants from the Northern Cape had sutherlandioside B and C [6]. These variations may possibly explain anti-cancer activity exhibited by extracts from different regions.

The present study shows a strong correlation between phenolics and flavonoids and the anti-oxidant activity, likewise, several studies have demonstrated a positive relationship between total content of phenolics and respective anti-oxidant capabilities of medicinal plants [20,21,22]. The plants referred to here as the Gansbaai 1 and Pearly Beach set possessed the highest concentrations of phenolics and flavonoids and consequently, these plants displayed the highest radical scavenging activity and reducing anti-oxidant activity. However, this is inconsistent with the observations linked to the Burgersdorp chemotype which consistently had the lowest concentrations of both phenolics and flavonoids, and interestingly, these plants did not turn out to display the lowest anti-oxidant power in both DPPH^•^ and FRAP methods, applied in this study (Figure 2 and Figure 3).

Despite variations that are site-specific, these data are in agreement with previous reports that have investigated the anti-oxidant activity of *S. frutescens* [10,11,12,13]. It is important to note that specific chemotypes have not been directly correlated to their anti-oxidant power in most studies in the past as their aims have not been placed on studying the bioactivity of plants collected from a range of localities and biogeographic considerations that may lead to different behaviors of extracts in pharmacological tests are not the main concern. However, extracts of *S. frutescens* have been shown to scavenge other reactive oxygen species that include superoxide ions, hydrogen peroxide, and nitric oxide, which are known to be involved in the pathogenesis of various health conditions [10,11]. Extracts are also known to have iron chelating capacity which is an important anti-oxidant feature [13]. It is thus likely that the anti-oxidant effects are linked to a complex mechanism that has diverse targets. For example, the suppression of the production of reactive oxygen species in various cell cultures such as neurons and microglial cells has been demonstrated by Jiang et al. [12] and this plays a protective role against oxidative stress linked to *S. frutescens*.

The Karoo (a semi-dessert region) and Western Cape (a coastal region with a Mediterranean climate) present with different climates that influence plant metabolism, leading to different chemicals occurring in plant extracts derived from different locations and the work of Zonyane et al. [6] indicate differential display of sutherlandins and sutherlandiosides for populations found at these localities, linking the habitat to the geospatial climatic conditions that the plants face. As we had collected the plants during November to December when the temperatures start to rise and the rain becomes less frequent in both the Karoo and the Western Cape, this in itself is likely to influence the availability of bioactives as many other authors discuss the impact of seasons on secondary plant metabolism is well established. Seasons and plant metabolism show remarkable biochemical adjustments to environmental changes and such changes need to be monitored for *S. frutescens* in the future.

The coastal Western Cape plants are subjected to wetter environments during the winter with rainfall levels being higher than those plants in the Karoo. Both temperature and moisture content are well known to cause changes to gene expression that modifies the biochemistry of plants at different degrees within a plant species occurring in a population. Harsher environments where a variety of stresses may exist lead to the de novo production of various metabolites that are intricately linked to the stress response allowing plants to adjust as part of their coping mechanisms to counter ROS generated due to changing external stress stimuli and flavonoids and terpenes, for example, accumulate at high levels to counter light induced oxidative stress [23]. The regulation of metabolism is central to many plant developmental processes, underpinning many ways in which plants respond to the environment, effecting both short-term and long-term changes. In environments such as the Karoo, that have dry hot summers with high light intensity, plants under such conditions may inherently accumulate more chemicals that facilitate the coping mechanisms needed to withstand the ROS-mediated reactions. It is interesting to note that those plants that come from Colesburg (within the Karoo region) had the highest anti-proliferative activity (lowest IC_50_ value of 158.7 µg·mL^−1^) against tested cancer DLD-1 cell line.

This study is important because there is no documentation of anti-oxidant activities of *S. frutescens* attached to the collection of plants from varying biogeographical localities. Recorded anti-oxidant activity is not surprising as this feature is ubiquitous in plants because it leads to coping with a range of environmental stresses that plants face due to their sessile nature [24]. Those populations that are face with environmental stresses are prone to produce excessive levels of reactive oxygen species (ROS) [23,24]. At basal levels, ROS are in fact important for normal plant growth and development. However, at excessive levels, they react with cellular macromolecules (proteins, lipids, and/or DNA) causing oxidative stress and interruption of metabolic enzymes and these result to impaired cell functions [24]. The anti-oxidant potential is variable in plants from species to species and in different genotypes of the same species plus there is a positive correlation between anti-oxidant potentials in plants and stress tolerance that is linked to the function of both flavonoids and terpenes. From this set of experiments, it is clear that the chemotype is in fact an important consideration which should be aligned with products claiming to have good anti-oxidant power for phytopharmaceutical producers.

The cytotoxicity of *S. frutescens* on colon cancer cell lines that has been recorded here (Figure 3) is consistent with previous studies that have demonstrated anti-proliferative effects of *S. frutescens* on SNO esophageal [25] and CaCo2 cell lines [9] which are cell lines related to colon cancer. Literature has also demonstrated the cytotoxicity of *S. frutescens* derived extracts on other various cancer cell lines such as breast cancer cell lines [7,8,26,27], leukemia [27], and cervical cancer cell lines [28] and presence of both flavonoids and terpenoidal saponins in the extracts suggest that these chemicals may be responsible for the bioactivity [4]. Although chemical variability is exhibited by this species based on its biogeographic placement (Figure 4), as shown also by another study of Zonyane et al. [6], *S. frutescens* shows promise as an anti-cancer agent and influence of site-dependent bioactivity should be taken into consideration for standardization of active pharmaceutical ingredient, especially when the intention is to commercialize this type of activity for human consumption.

Differentiation in the biological properties of *S. frutescens* can also be attributed to variability in the chemical composition of *S. frutescens* specimens screened. The biological activities of *S. frutescens*, like any medicinal plant, are due to its complex chemical makeup [29,30]. Literature reports the presence of flavonoids and triterpene, apart from other compounds that are mainly generated via central (or primary) metabolism such as pinitol, canavanine, and a suite of amino acids [1,5,29]. The presence of these compounds in *Sutherlandia* is thought to be responsible, individually or in combination, for the broad-spectrum bioactivity displayed by the plant [5]. In this particular study, we tentatively identified several chemicals that are prominent in *S. frutescens* that have not been fully characterized but they belong to the flavonoid and terpene saponin groups. It is well accepted that both flavonoids and terpene saponins exhibit their effects in complex ways to combat the establishment and growth of cancers, inducing the expression of genes that induce apoptosis and interfere with cell cycle leading to its arrest [31,32]. Because sutherlandins are kaempferol- and quercetin-derived compounds, and such flavonoids do not only prevent cancer cell division and growth, but also, they interfere with cancer cell migration in vivo plus they downregulate gene expression involved in cancer cell progression [31,32]. With more research a clearer picture is starting to emerge in terms of the anti-cancer bioactivity of *S. frutescens* as different cell signaling mechanisms (i.e., inhibition of the Gli/Hh pathway and p13-kinase pathway), cytochrome-mediated apoptosis, cell cycle arrest, to name a few, are amongst the mechanisms that are prominent in the cancer inhibition effect [8,9,12,15,16,18]. Triterpenes are powerful in their free radical scavenging action and affect anti-oxidant enzymes, changing the route to pathological development inside cells [33] plus flavonoids and terpene saponins affect accumulation of anti-inflammation proteins [4,31,32]. It is also likely that sutherlandins and sutherlandiosides work in concert to illicit anti-oxidant and anti-cancer effects; and, that the other phenolic compounds also add to the synergism making the extracts more potent.

At this stage, it is unclear which chemicals in the extract possess the highest anti-cancer action although new information related to this aspect is starting to appear in the literature as fractions with high amounts of sutherlandioside D were recently found to be more potent against prostate cancer cells [17]. It would thus be interesting to further the tests on pure samples of sutherlandins and sutherlandioside in terms of their anti-oxidant and anti-cancer inhibition effects. There is considerable variation in other phenolic compounds that do not necessarily belong to the sutherlandin group, eluting between 6 and 7 min (Figure 1) that seem to occur in more abundance in the Gansbaai 1 and Pearly Beach populations and this may further explain the grouping of these plants in cluster 3 (Figure 5). However, there is growing evidence of phytochemical variation in *S. frutescens* plants found in diverse provinces as attested in related previous studies [5,6,30]. Therefore, variation in chemical ingredients of this plants is likely to have implications on its pharmacological properties and future bioactivity studies should strongly take this into account. Intraspecific chemotypic variation in other plant species is well documented to affect biological activity of plant-derived medicinal preparations, and for this reason, chemotype selection is a critical step in quality assured and standardized medicinal plant extracts [30].

*Sutherlandia frutescens* has been extensively studied for its various biological activities without consideration of the plant’s heterogeneity of its chemical composition which is likely to have bearing on efficacy. The findings of this study highlight the importance of quality control of this medicinal plant as specimens demonstrated variability in their biological activities concerning anti-oxidant and anti-cancer activities. It is important to note that *S. frutescens* also has other biological activities in addition to anti-cancer and anti-oxidant activities, therefore the variability recorded in this study warrants further investigation on the chemical variants for efficacy in other biological activities that are associated with this medicinal plant species. A full-scale characterization of the extracts of *S. frutescens* is still urgently needed as there are many chemicals that occur in the plant that remain unknown (Table 3). With aims to utilize extracts of *S. frutescens* as a potential phytotherapy against cancer, it becomes more important to choose chemotypes from wild collected materials that exhibit the greatest biological activity whose chemistry is better defined. Such a study will assist the industry that has commercialized the extracts of *S. frutescens* to earmark those chemotypes that have been shown to possess the most relevant bioactivity.

## 5. Conclusions

A strong relationship between ecotypes and their respective anti-oxidant and anti-cancer activities, which is likely due to variable phenolic and flavonoids composition of samples from different areas, is indicated in this study. Although the anti-oxidant activity could not be used as an indicator of potency against cancer cells tested, we could pinpoint those samples that illustrated high bioactivities for both aspects. We thus show the Western Cape plants as being a better source of anti-oxidant material but the Northern Cape are superior as anti-cancer agents against the tested cell line. This allows a more informed choice in terms of those elite specimens of *S. frutescens* that should be tested in the future in in vivo studies, to determine whether the data generated herein is reflected in a more clinical setting. The variable bioactivity confirmed in this study, supports the need to add to already existing cytotoxicity analysis of *S. frutescens* but future studies should focus on individual chemotypes to better understand the influence of different localities on their toxicity profiles. Such information is important for guiding various industries that manufacture products of this particular species.

## Figures and Tables

**Figure 1 antioxidants-09-00152-f001:**
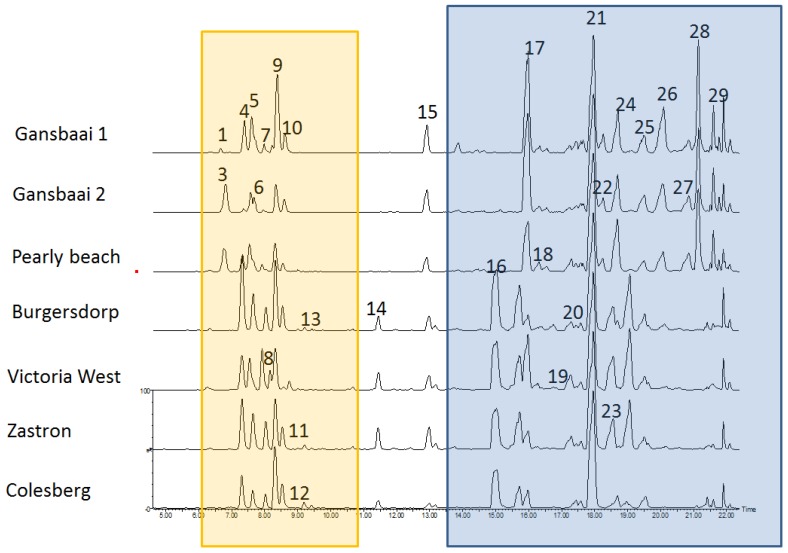
Examples of chromatograms of *Sutherlandia frutescens* extracts prepared from plants occurring in different localities. The yellow region represents the sutherlandins and the blue region the sutherlandiosides.

**Figure 2 antioxidants-09-00152-f002:**
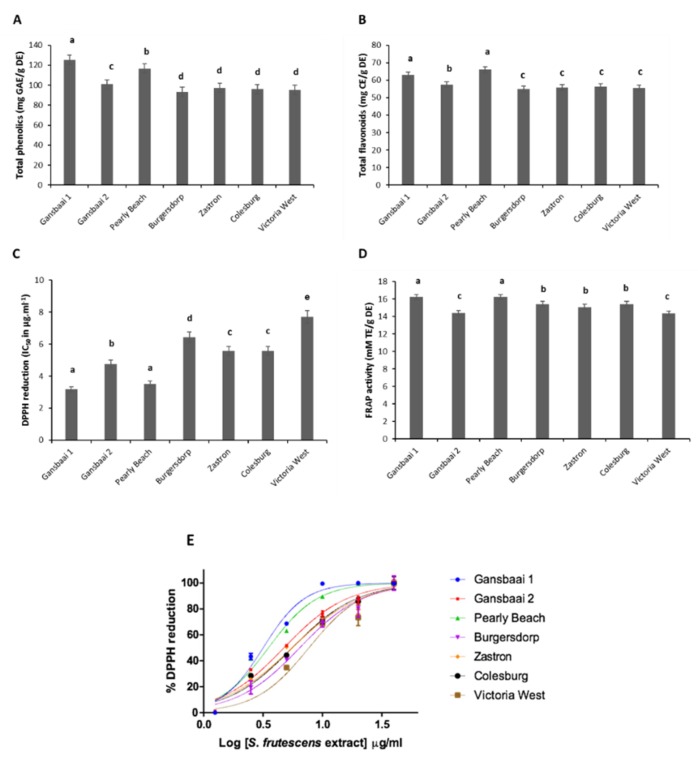
Anti-oxidant effect of Sutherlandia frutescens plants from different localities studied using in vitro tests. (**A**) Total phenolic content, (**B**) Total flavonoid content, (**C**) comparative DPPH free radical scavenging activity (IC50) and (**D**) ferric reducing anti-oxidant power (Trolox equivalent) and IC50curve for free radical (DPPH^•^) scavenging activity (**E**). Mean values (±SD) of triplicate measurements are presented. Different letters above average bars denote statistically significant differences at *p* < 0.05.

**Figure 3 antioxidants-09-00152-f003:**
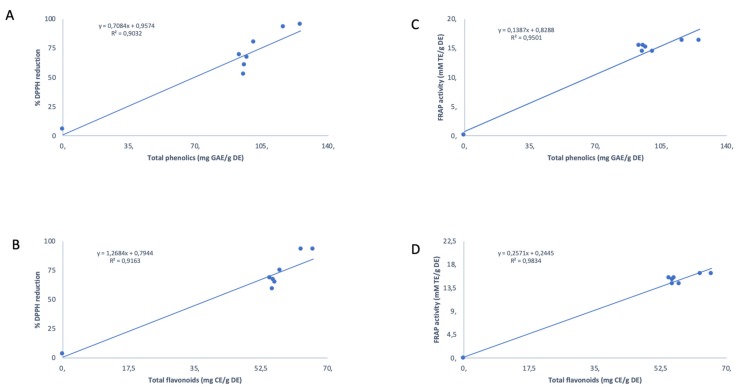
Correlation between the studied anti-oxidant phytochemicals (total phenolics and total flavonoids) and anti-oxidant capacity of *Sutherlandia frutescens* plants from various geographic locations. (**A**) %DPPH versus total phenolic content (**B**) FRAP activity versus total phenolic content (**C**) %DPPH versus total flavonoid content (**D**) FRAP activity versus total flavonoid content.

**Figure 4 antioxidants-09-00152-f004:**
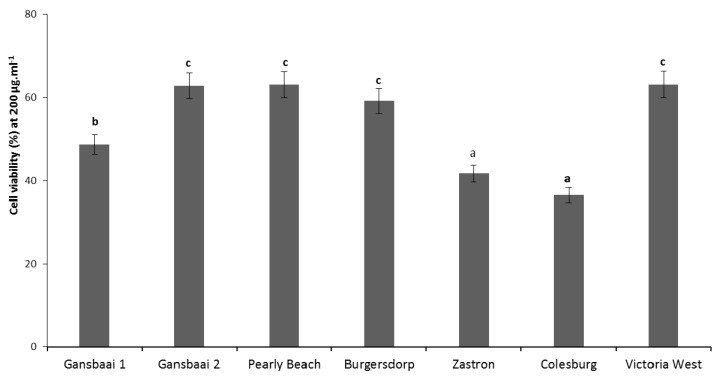
The cell viability of DLD-1 colon cancer cell lines treated with *Sutherlandia frutescens* extracts at 200 µg·mL^−1^ from different geographic localities using an in vitro test. Values are reported as percentages and different letters above bars denote statistically significant mean differences at *p* < 0.05.

**Figure 5 antioxidants-09-00152-f005:**
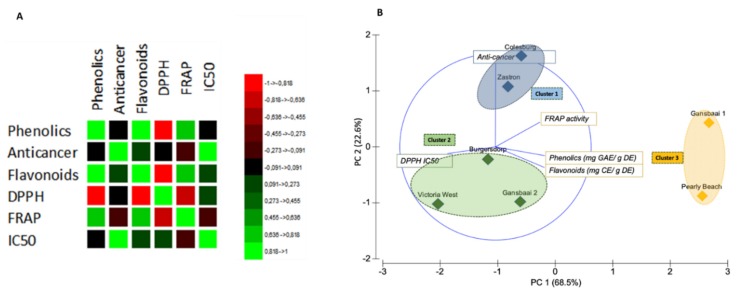
Relationship of tested bioactivities from different localities (**A**) Correlation matrix based on Pearson’s coefficient scores showing correlated pairs amongst test parameters (*p* = 0.05). Green indicates positive correlations and negative correlations are shown in red and the intensity of the colour is proportional to the correlation coefficients. (**B**) Principal component analysis indicating tested bioactivities and origin of plant extracts of *Sutherlandia frutescens*. Extracts from different sources are identified by the locality (represents active observations) and clusters are based on tested attributes of *S. frutescens* extracts (represents active variables).

**Table 1 antioxidants-09-00152-t001:** *Sutherlandia frutescens* specimens from different habitats of South Africa and chemical compounds that are important for their chemotypic differences.

Province	Geographic Location	Quercetin and Kaempferol Derived Flavonoids	Terpenoid Saponins (Cycloartanol Glycosides)
**Western Cape**	Gansbaai 1 34°32′09.4″S 19°24′26.0″E	Sutherlandin B 7.66_755.2021 *	18.69_737.4104 *
	Gansbaai 2 34°32′34.8″S 19°24′38.6″E	Sutherlandin B 7.66_755.2021 *	18.69_737.4104 *
	Pearly Beach 34°40′47.3″S 19°33′20.0″E	Sutherlandin B 7.66_755.2021 *	18.69_737.4104 *
**Eastern Cape**	Burgersdorp 31°02′26.0″S 25°44′08.0″E	Sutherlandin A Sutherlandin B	Sutherlandioside A Sutherlandioside B
**Free State**	Zastron 30°29′49″S 27°09′71″E	8.73_593.1501 * 7.99_609.1448 *	Sutherlandioside A Sutherlandioside B
**Northern Cape**	Colesberg 30°48′17.3″S 24°58′52.4″E	Sutherlandin A Sutherlandin C Sutherlandin D	Sutherlandioside A Sutherlandioside B
	Victoria West 31°32′48.2″S 23°35′00.4″E	Sutherlandin A Sutherlandin C Sutherlandin D	Sutherlandioside B Sutherlandioside C

* Elution time and mass number of unidentified compound.

**Table 2 antioxidants-09-00152-t002:** IC_50_ values of *Sutherlandia frutescens* extracts produced from plants growing at different geographic localities in South Africa.

Plant Type	Provincial Location	IC_50_ (µg·mL^−1^)
**Gansbaai 1**	**Western Cape**	**176.7**
Gansbaai 2	Western Cape	>200
Pearly Beach	Western Cape	>200
Burgersdorp	Eastern Cape	>200
**Zastron**	**Free State**	**172.7**
**Colesburg**	**Northern Cape**	**158.7**
Victoria West	Northern Cape	>200

Boldtype face highlights extracts that are regarded as being active.

**Table 3 antioxidants-09-00152-t003:** Constituents tentatively identified in *Sutherlandia frutescens* leaf extracts.

	RT (min)	[M + H]^+^ Found	MS^E^ Fragment Ions	Elemental Formula	UV Max	Tentative Identity
1	6.31	903.2404	**303.051**, 465.100, 597.142, 771.187	C_38_H_46_O_25_	256; 347	Quercetin glycoside (Sutherlandin A/B derivative)
2	6.66	427.1911	149.069, 287.059, 177.07, 120.091, 225.163, 207.126, 163.056, 303.059, 387.197, **427.194**	C_28_H_26_O_4_	245; 323	Unknown flavonoid
3	6.80	771.1998	**303.051**, 609.143, 771.198	C_33_H_38_O_21_	256; 347	Quercetin-glycoside (Sutherlandin A/B derivative)
4	7.32	741.1874	**303.05**, 609.144, 741.186, 287.057, 127.041	C_32_H_36_O_20_	256; 352	Sutherlandin A
5	7.54	741.1855	**303.05**, 609.148, 741.19, 187.06, 127.039	C_32_H_36_O_20_	255; 354	Sutherlandin B
6	7.66	755.2021	**287.055**, 755.200, 777.1755	C_33_H_38_O_20_	265; 351	Kaempferol glycoside (Sutherlandin C/D derivative)
7	7.99	609.1469	**303.051**, 609.107, 287.059, 187.065	C_27_H_28_O_16_	256; 347	Quercetin glycoside (Sutherlandin A/B derivative)
8	8.16	725.1927	**287.055**, 725.211, 593.146, 303.035, 187.061, 593.148	C_32_H_36_O_19_	264; 347	Sutherlandin C isomer (small peak)
9	8.32	725.1920	**287.055**, 593.149, 725.192, 127.039, 187.06	C_32_H_36_O_19_	265; 347	Sutherlandin C
10	8.53	725.1899	**287.055**, 593.15, 725.191, 127.042, 187.049	C_32_H_36_O_19_	266; 348	Sutherlandin D
11	8.73	593.1501	**287.055**, 593.1500, 615.130	C_27_H_28_O_15_	265,349	Kaempferol glycoside (Sutherlandin C/D derivative)
12	9.01	1079.2913	**177.056**, **303.048**, 641.152, 1079.287, 947.248, 145.035	C_66_H_45_O_15_	255; 347	Quercetin glycoside (Sutherlandin A/B derivative)
13	9.21	593.1508	**287.055**, 593.150, 615.134	C_27_H_28_O_15_	265; 347	Kaempferol glycoside (Sutherlandin C/D derivative)
14	11.79	829.4537	**505.348**, 487.337, 177.067, 846.464, 851.438	C_49_H_64_O_11_	None	Unknown triterpenoid
15	12.90	831.4672	**505.353**, 487.342, 689.387, 203.152, 471.349, 853.458, 705.363	C_49_H_66_O_11_	None	Similar to compound 505 at RT 11.71, Unknown triterpenoid
16	15.02	653.4250	**473.363**, 455.353, 437.342, 491.373, 419.331, 635.416, 653.427, 675.408	C_36_H_60_O_10_	None	Sutherlandioside A
17	15.97	813.4630	**489.358**, 471.347, 651.410, 813.461, 830.489, 835.447	C_42_H_68_O_15_	None	Cycloartanol glycoside
18	16.31	899.4630	**489.358**, 471.347, 916.491, 719.400, 657.397, 453.336, 921.447	C_45_H_70_O_18_	None	Unknown cycloartanol glycoside
19	17.3	653.4252	**437.341**, 455.352, 419.330, 489.357, 617.404, 635.415, 675.407, 653.424	C_36_H_60_O_10_	None	Sutherlandioside A isomer
20	17.57	809.4454	**489.357**, 471.347, 177.06, 437.339, 827.457	C_46_H_65_O_12_	None	Cycloartanol glycoside
21	17.97	651.4094	**489.358**, 471.346, 453.336, 668.437, 873.392	C_36_H_58_O_10_	None	Sutherlandioside C
22	18.21	825.4257	**437.341**, 419.331, 455.353, 541.355, 789.407, 771.396, 807.414, 523.343, 842.455, 848.409	C_42_H_65_O_16_	None	Cycloartanol glycoside
23	18.53	653.4268	**437.342**, 455.353, 473.363, 489.358, 419.33, 635.416, 653.426, 675.408	C_36_H_60_O_10_	None	Sutherlandioside B
24	18.69	737.4104	**489.359**, 471.348, 453.338, 737.412, 719.401, 657.402, 701.391, 759.395	C_39_H_60_O_13_	None	Cycloartanol glycoside
25	19.49	635.4160	**455.353**, **437.342**, 419.333, 657.398, 473.364635.419	C_36_H_58_O_9_	None	Sutherlandioside D
26	19.99	721.4149	**437.341**, 455.352, 703.404, 419.330, 685.393, 667.382, 641.407, 721.415, 743.397	C_39_H_60_O_12_	None	Cycloartanol glycoside
27	20.84	635.4144	**437.341**, 455.351, 419.330, 657.396, 489.358, 473.362, 635.414	C_36_H_58_O_9_	None	Sutherlandioside D isomer
28	21.13	721.4154	**437.341**, 455.352, 721.416, 419.331, 703.404, 473.362, 685.392, 738.443	C_39_H_60_O_12_	None	Cycloartanol glycoside
29	21.61	703.4058	703.406, **437.342**, 455.352, 419.332, 229.160, 685.396	C_39_H_58_O_11_	None	Cycloartanol glycoside

MS^E^ fragments in boldtype face refers to the base peak (highest peak) [5].

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
