# Peer review of "The Implication of Chemotypic Variation on the Anti-Oxidant and Anti-Cancer Activities of Sutherlandia frutescens (L.) R.Br. (Fabaceae) from Different Geographic Locations"

_antioxidants, 2020, doi:10.3390/antiox9020152_

Round 1
Reviewer 1 Report
The article (ID: antioxidants-710854) titled: The implication of chemotypic variation on the antioxidant and anti-cancer activities of Sutherlandia frutescens (L.) R. Br. (Fabaceae) from different geographic locations.
This manuscript presents different extracts of Sutherlandia frutescens (cancer bush) that contain flavonoids and polyphenols likewise other chemical constituents. The composition depends on the origin region where the bushes were cultivated. The authors determine the composition of chemical from different regions and present the bioactivity of the Sutherlandia frutescens extracts using in vitro antioxidant and anticancer assays. They conclude that Gansbaai and Pearly Beach (Western Cape) showed superior antioxidant activity. In addition, all the extracts at 200 µg.ml-1 were cytotoxic for DLD-1 colon cancer cells exhibiting Colesburg (Northern Cape) the highest activity. These results confirm that S. frutescens extracts display variability in their bioactive capacity based on their natural location.
As such, the manuscript is worthy of publication, but some issues require attention before it is acceptable.
I would recommend to re-write clearer the introduction.
See the following mistakes;
Line 126. Explain the conditions of the ultra-sound sonication. Was it made in cool? Line 193. I think 20 µl is a very low volume. Is this volume correct? Figure 2. Align figure 2C with 2D going up figure 2C in addition, the title of Pearly Beach, Burgersdorp and Victoria West from figure 2C have been lost. Figure 2. It seems to be written Burger sdorp instead of Burgersdorp, in the x axis. Figure 2D. In y axis, it is missed the final parenthesis. Reference 1. Review the title, there is a superscript. Reference 10. The title has been lost, please write in the correct position: The antioxidant potential of Sutherlandia frutescens. Reference 12. That is wrong. The year 2014 is twice, before the title and after the volume. Delete 2014 before the title. Reference 16. Review, the name of the authors: Gelderblom, W. C. Reference 19. There are two references mixed. The number 20 is duplicated with number 19. Please, eliminate. Reference 25. Change Cronje by Cronjé. Reference 30 and 31. Correct them; authors, year… Reference 32. Correct the pages: 178-18 to 178-181. Also, correct the year is twice. Please, review all the references, some of them have capital letters in all the words of the title and others references not. Unify. Before the title, it is sometimes used a full stop and others a comma or semicolon. Please, unify.Author Response
Introduction – We have rewritten sections of the introduction and this is hoped that it is much clearer. All new sentences and re-written sections are indicated in red.
Line 137 – A sentence has been added where we have indicated that the ultra-sound sonication was undertaken under cool conditions and we did this by physically adding some ice to the sonication bath water.
Line 227 to 228 – We contacted the company and consulted with Dr Jiangho Wu (Reaction Biology Corporation) who conducted this assay. He assured us that they use 20 microliters for the assay in a 384-plate and that it was sealed so that the medium would not evaporate.
We have included this new information; and yes, the value of 20 microliters is correct. This value is the appropriate amount for the CellGlo assay used by Reaction Biology. We have also indicated that this part of the test was conducted at Reaction Biology Corporation in the USA.
References:
All the references have been corrected as per suggestion and we have used a reference manager to conduct these changes plus we have printed out the references and made sure that they are all uniform. Added references are indicated in red.
Reviewer 2 Report
The authors of the manuscript “The implication of chemotypic variation on the anti- oxidant and anti-cancer activities of Sutherlandia frutescens (L.) R.Br. (Fabaceae) from different geographic locations” described the study of different population of Sutherlandia frutescens and their antioxidant and anticancer activities .
The authors in the introduction failed to describe the secondary metabolites found in S. frutescens, also a poor correlation of why are you evaluating these activities, is it based on the traditional use? .more discussion about its traditional use. I
In the text , I missed the chemical structures, in the discussion it is hard to understand why you mainly focus in the flavonoids, seems based on the figure 1, that there is a way more variation in the saponins in the different ecotypes, or maybe other compounds that you did not isolate or identified. Saponins are well known anticancer activity agents I missed more discussion about the anticancer and antioxidant activities of the flavonoids and how merge that.
The manuscript has the quality and could be accepted after minor changes
Author Response
Introduction – We have provided a description of the secondary metabolites found in S. frutescens and specifically focused on the sutherlandins and sutherlandiosides that are the dominant chemicals of these plants (refer to Line 72 to 81). All new sections are written in red.
A section on the traditional uses of this species is now included (Line 90 to 94).
Justification and correlation of why the specific activities are being studied has been provided.
Refer to Line 49 to 64 where we provide a general description of flavonoids in plants and how they contribute to antioxidant activity and how this can prevent cancers.
Refer to Line 65 to 71 where we provide a description of terpene saponins and how these chemicals function as anticancer agents. We describe the sutherlandins (flavonoids) and sutherlandiosides (terpene saponins)and provide chemical structures in a supplementary document.
We have rewritten information on previous studies on anticancer activity and specifically focused on papers that show mode of action of extracts (refer to Line 95 to 97). We have also made a direct correlation of why we evaluated antioxidant activity and anticancer activity and this is clearly articulated and is linked to the fact that sutherlandins are flavonoid and sutherlandiosides are terpene saponins and both these compound groups are well established as having these types of activities.
We have now shown the importance of the terpene saponins as well and this indicated in line 486 to 498. The latter section deals directly with the power of triterpene saponins as antioxidants (lines 494 to 496).
We value the comments and we hope that this manuscript reads a lot better, with better flow and it is now acceptable for publication.